# Antimicrobial Peptides: Avant-Garde Antifungal Agents to Fight against Medically Important *Candida* Species

**DOI:** 10.3390/pharmaceutics15030789

**Published:** 2023-02-27

**Authors:** Gina P. Rodríguez-Castaño, Frank Rosenau, Ludger Ständker, Carolina Firacative

**Affiliations:** 1Studies in Translational Microbiology and Emerging Diseases (MICROS) Research Group, School of Medicine and Health Sciences, Universidad de Rosario, Bogota 111221, Colombia; 2Institute of Pharmaceutical Biotechnology, Ulm University, 89081 Ulm, Germany; 3Core Facility for Functional Peptidomics, Faculty of Medicine, Ulm University, 89081 Ulm, Germany

**Keywords:** antifungals, antimicrobial peptides, *Candida*, candidiasis, mycosis

## Abstract

Expanding the antifungal drug arsenal for treating *Candida* infections is crucial in this era of the rising life expectancy of patients with immunosuppression and comorbidities. Infections caused by *Candida* species are on the rise, including those caused by multidrug-resistant strains or species, and the list of antifungals approved for the treatment of these infections is still limited. Antimicrobial peptides (AMPs) are short cationic polypeptides whose antimicrobial activity is under intense investigation. In this review, we present a comprehensive summary of the AMPs with anti-*Candida* activity that have undergone successful preclinical or clinical trials. Their source, mode of action, and animal model of infection (or clinical trial) are presented. In addition, as some of these AMPs have been tested in combination therapy, the advantages of this approach, as well as the studied cases that have used AMPs and other drugs concomitantly to fight *Candida* infections, are described.

## 1. Introduction

*Candida* species are often part of the normal microbiota of humans. They can inhabit the skin, the vaginal tract, and the gastrointestinal tract from the oral cavity to the anus [1]. In these niches, *Candida* must survive host factors that shape the microbiota and compete against other microorganisms. Epithelial cells, for instance, which in most cases are the first host cells to be in direct contact with the fungal cells, secrete a plethora of antimicrobial agents, such as antimicrobial peptides (AMPs), and various digestive enzymes, which have antifungal activity and restrain *Candida* from the invasion of the underlying tissue [2]. If some of these factors are impaired, *Candida* can invade and cause cutaneous, mucosal, and systemic infections, most of which are seeded by commensal yeasts that inhabit the body [3]. In addition, hospital personnel, devices and infusates can also be the source of infection [4]. Once in the host, *Candida* must face a complex interplay between the expression of fungal virulence factors, including adherence, invasion, and cell damage, and the host immune system, which normally responds by activating antimicrobial activities and killing the yeast [5]. Among the risk factors that predispose the host to these infections are the use of antibiotics, which relieve *Candida* from bacterial competition, the disruption of anatomical barriers, and a weakened immune system [1,6]. Indeed, in recent decades, the rising number of immunocompromised patients, their longer life expectancy, the more extensive use of indwelling medical devices and antibiotics, among other factors, have given rise to an increased number of infections caused by *Candida* species [7,8]. It is estimated that there are more than 400,000 life-threatening *Candida* infections each year, with a mortality rate between 46% and 75% [9,10].

*Candida albicans* is the main etiologic agent of these infections globally. The ARTEMIS Global Antifungal Surveillance Program estimated that, in a 6.5-year study, this species accounted for 66% of the studied cases. However, during the time span of the study, the isolation rate of this species had a 10% overall decrease. Among the less common *Candida* species, *Candida glabrata*, *Candida tropicalis*, *Candida parapsilosis*, and *Candida krusei* had substantial incidences, accounting for 11, 6, 5, and 2% of the total cases, respectively. It was also noted that *C. tropicalis* and *C. parapsilosis* had about a 3% increase each in their isolation rate during the study [11]. More recently, *C. auris*, first isolated in 2009 in Japan, has been recognized by the Centers for Disease Control and Prevention (CDC) as an emerging multidrug-resistant fungus that is rapidly spreading globally and presents a serious global health threat [12].

Most antifungals approved for the treatment of *Candida* infections belong to three main classes: echinocandins, polyenes, and azoles [13]. These drugs have different mechanisms of action. The most novel class, echinocandins, are cyclic peptides that disrupt the fungal cell wall by targeting (1,3)-β-d-glucan synthase. The inhibition of the synthesis of (1,3)-β-d-glucan, an essential component of the yeast cell wall, has both fungistatic and fungicidal effects. Cell wall synthesis is blocked and eventually its integrity is affected, leading to its destruction. Echinocandins are specific to fungal cells, and thus they have an excellent safety profile and their interaction with other drugs is low [14]. Another key component of the fungal cell membrane is ergosterol. This is targeted by the most commonly used polyene, amphotericin B, which forms aggregates on the membrane and functions like a fungicidal “ergosterol sponge”. However, amphotericin B has more serious side-effects, including nephrotoxicity, and has poor oral bioavailability [15]; as such, for the treatment of *Candida* infections, amphotericin B is mostly used when other drug classes are not effective. Meanwhile, azoles, such as fluconazole and voriconazole, do not act directly on ergosterol but block a key enzyme in the ergosterol biosynthetic pathway, the cytochrome P450 lanosterol 14α-demethylase [16]. These drugs have excellent oral bioavailability but have a high potential for interacting with other drugs [17]. Another shortcoming of azoles is their fungistatic effect, which indicates that they only inhibit the growth of the pathogen, and it has been observed that yeast cells can accumulate mutations in a non-dividing state [18]. This promotes the development of antifungal drug resistance, which is a serious public health threat. According to the CDC, there are almost 35,000 cases of an infection caused by drug-resistant *Candida* species each year, from which 1700 deaths occur each year in the USA alone [19].

Resistance to echinocandins can be achieved through increased levels of chitin in the cell wall, through mutations in the catalytic subunit of glucan synthase, and via the prolonged drug exposure of the isolates [20,21]. Lately, *C. albicans, C. glabrata* and *C. parapsilosis* isolates, which are resistant to echinocandins, have emerged in both laboratory and clinical settings [22]. Resistance to amphotericin B, which is rare in *Candida* species, can happen when enzymes in the ergosterol biosynthesis pathway accumulate mutations or can occur via the use of azoles that reduce ergosterol cellular levels [23]. In another ARTEMIS global surveillance study, it was estimated that the resistance to the azoles fluconazole and voriconazole was 1% for both antifungals in *C. albicans*, 17% and 10% in *C*. *glabrata*, 4% and 5% in *C. tropicalis*, 3% and 2% in *C. parapsilosis*, and 74% and 7% in *C. krusei*, respectively [24]. The multi-drug resistant *C. auris* is commonly resistant to fluconazole and amphotericin B, and, although resistance to echinocandins is rare, a low number of isolates (about 4%) show resistance to all classes of antifungals [22,25]. The emergence of resistance poses severe limitations to the efficiency of antifungal therapies. Therefore, the development of new drugs with antimicrobial properties is urgently needed. A broad spectrum of AMPs is being evaluated for this purpose, including the already commercially available echinocandins mentioned above.

In this review, we have elaborated a comprehensive list of non-commercial AMPs that have been evaluated in preclinical and clinical trials and were observed to have low toxic effects. The included AMPs have been the subject of preclinical or clinical studies against infections caused by *Candida* spp., or of preclinical or clinical studies for the treatment of fungal infections caused by other pathogens but that have been proven to have in vitro activity against *Candida* species (Table 1). The AMPs shown here can be used as standalone drugs, and this is the starting point for evaluating their mode of action and efficiency; however, we also emphasized how AMPs can be used in combination therapies and that they have shown synergy with other antimicrobial agents. This strategy can reduce the amount of each drug needed, thus reducing the side effects, increasing the efficiency of conventional drugs in resistant strains, broadening the antimicrobial spectrum and, as mentioned before, preventing the emergence of drug resistance if both drugs have different mechanisms of action, including their recognition of membrane components.

## 2. Characteristics of Antimicrobial Peptides

Antimicrobial peptides are small, mostly cationic polypeptides found in all organisms. More often they contain between 2 to 50 amino acids and have a net positive charge of +2 to +7 [26]. The Apd3 Antimicrobial Peptide Database estimates that there are over 3000 natural AMPs with known antimicrobial activity [27]. One common characteristic of AMPs is that they have amphiphilic properties: one section is hydrophobic while the other is charged. Cationic AMPs are attracted to anionic membranes through electrostatic interactions, meaning that they are mostly attracted to the microbial cell membranes that are abundant in negative charges, while the surface of the membranes of plants and animals has no net charge [28]. The mere interaction of peptides with the lipid bilayer is not enough to cause cell destruction; the assembly of peptide monomers is required for membrane permeabilization and thus non-assembled peptides can avidly bind the membrane but distribute randomly on the membrane surface and not cause membrane destabilization [29]. The first interaction of AMPs with the lipid bilayer varies depending on the nature of the peptide and the composition of the membrane, but a general mechanism is explained by the Shai–Matsuzaki–Huang model. The positive charged side of the peptide binds to negatively charged lipid headgroups, then the peptide integration causes the thinning of the outer half of the membrane. The membrane crossing of the peptide is also explained by different models: the barrel-stave model, the carpet model, the toroidal pore model, and the micellar aggregate channel model [30]. Antimicrobial peptides can also modulate the immune response of the host to the pathogen by binding to inflammatory components such as lipopolysaccharides (LPS), which can avoid endotoxin shock; this has a high rate of mortality and no effective treatment [31]. Among the recognized antifungal mechanisms of AMPs is the interaction with the fungal membrane and the cell wall. Some targets of the anti-*Candida* peptides that are currently known, including membrane binding and the interaction and binding to the cell wall components, such as mannan, (1,3)-β-d-glucan and chitin, are schematically depicted (Figure 1).

Antimicrobial peptides have many advantages over conventional antifungals. One is that there is a substantial source of these molecules in nature. In addition, for the majority of classes, the diversity is so large that the same peptide is rarely recovered from two different species even if they are closely related [32], and a single organism can have more than 24 different AMPs [26]. Another advantage is that they can have a broad spectrum of antimicrobial activity; for instance, it is estimated that 853 peptides possess antibacterial and antifungal activities, 65 have antibacterial, antifungal, and antiviral activities, and 9 have antibacterial, antifungal, antiviral, and antiparasitic activities [33]. Lastly, resistance to these molecules is curtailed for various reasons. One is that they act faster than antibiotics, resulting in fewer generations of the pathogen, and thus making it less likely that the microorganism acquires a beneficial mutation. Another reason is that they usually act on more than one target, and that the modification of the physicochemical properties of the membrane in order to impede the interaction with AMPs has a high evolutionary cost; thus, it is more difficult for a pathogen to counteract their effects by point mutations. However, some experiments on the evolution of resistance to peptides have observed a decreased susceptibility to AMPs via mutations in a single gene or in membrane-related resistance mechanisms, for example, a reduced cell wall with a negative charge through mutations in different genes [20,34,35]. Nonetheless, the natural production of AMPs in organisms is less likely to prompt resistance because when exposed to an infection, hosts use a synergistic mixture of various AMPs that probably have different mechanisms of action [36]. All these characteristics make AMPs good candidates for drug development. One disadvantage is that during the development phase, high cytotoxicity against host cells can come to light [37].

## 3. Antimicrobial Peptides in Preclinical and Clinical Trials

Antimicrobial peptides can be found across all kingdoms of life, although, currently, most known AMPs belong to the animal kingdom (77%). Nevertheless, not only from vertebrates and invertebrate animals but from other kingdoms, several AMPs have been evaluated as alternative agents to treat *Candida* infections. The different AMPs with anti-*Candida* activity that have been evaluated in different animal disease models are summarized in a concise manner (Table 2). Depending on whether the antifungal properties of the AMPs have been the subject of intensive study, such as clinical trials, we classified and described these as major therapeutically relevant classes of AMPs (Table 3).

### 3.1. Antimicrobial Peptides from Vertebrates

#### 3.1.1. Antimicrobial Peptides from Mammals

In mammals, secretory fluids, such as milk, saliva, tears, and nasal secretions, as well as cells of the innate immune system and epithelial cells, are sources of a large diversity of AMPs that belong mainly to three families: defensins, cathelicidins, and histatins [33].

**Defensins**. Two AMPs that belong to the family of defensins are RTD-1 and Bac8c. **RTD-1** is an 18-mer cyclic peptide stabilized by three disulfide bonds found in the leukocytes of rhesus macaques. It belongs to a type of defensin, θ-defensins, that are only found in Old World monkeys. RTD-1 has been shown to be effective in preclinical models of bacterial sepsis, SARS-CoV-2 infection, *Pseudomonas aeruginosa-*induced cystic fibrosis, endotoxin-induced lung injury and in a model of systemic candidiasis using *C. albicans*. In this last study, different routes of administration of RTD-1, intravenous, subcutaneous, or intraperitoneal, promoted the clearance of the pathogen, modulated systemic inflammation, and was non-toxic to mice. RTD-1 is also highly stable in serum and plasma, resistant to proteases, and non-immunogenic. In vitro RTD-1 is effective in killing both planktonic cells and biofilms of drug-sensitive and multidrug-resistant *C. albicans* [62]. **Bac8c** is an 8-mer derivative of Bac2A, which was modified from bactenecin, a host defense peptide isolated from bovine neutrophils. Bac8c has antibacterial and antifungal activity. In vitro studies on *C. albicans* evidenced that Bac8c induces disturbances in the membrane potential, generates pores and membrane permeabilization, suppresses cell wall regeneration, disrupts electron transport, and finally causes cell death; the depolarization of the membrane occurs within 5 min. Another advantage of Bac8c is its short sequence (Table 1), which ensures a low-cost production [63]. An in vivo study used a variant of Bac8c with the D-amino acid D-Bac8c^2,5Leu^. This study investigated the effectiveness of the peptide in treating a bacterial infection when applied as catheter lock solutions in a rat central venous catheter infection model. D-Bac8c^2,5Leu^ was effective and displayed low immunogenicity and low cytotoxicity [64].

**Cathelicidins**. In the family of cathelicidins, the peptides LL-37, KR-12-a5, K9CATH, ZY13, indolicidin, omiganan, protegrin-I, and iseganan, are found. **LL-37** (37-mer) results from the proteolytic cleavage of human CAP18, and although it is linear, it acquires an α-helical conformation upon contact with the lipid bilayer. In *Candida* cells, it prefers to bind to mannan, and partially to chitin or glucan. It disrupts the lipid bilayer via a toroidal pore mechanism and changes the organization of the membrane by reducing the levels of glucan and mannan in the cell wall, exposing (1,3)-β-d-glucan, and creating ergosterol-dense and ergosterol-free areas. This phase separation of the membrane ultimately leads to membrane permeabilization [65,66]. In a murine urinary tract model, where the peptide and *C. albicans* were mixed and then injected into the urinary tract, LL-37 prevented the adhesion of the yeast to the bladder by more than 70% [59]. LL-37 also has also been the subject of clinical trials for other conditions; one study measures the efficacy of LL-37 cream on diabetic foot ulcers (NCT04098562) and another one, which was completed in 2021, evaluates intratumoral injections of LL-37 for melanoma (NCT02225366). **KR-12-a5** is a 12-mer α-helical peptide derived from LL-37 with both antimicrobial and anti-inflammatory activities that are maintained in physiological salts and human serum. Its antimicrobial activity is exerted by permeabilizing the cell membrane, which damages its integrity. It is independent of charge, but LPS neutralization does require a net positive charge and a balance of hydrophobicity [67,68]. KR-12-a5 shows antimicrobial activity in vitro against several oral pathogens, including *C. albicans* [69]. In vivo, it has been evaluated for other conditions; one study examined the reversal of the adverse effects of a LPS treatment on osteogenic differentiation in mice, in which KR-12-a5 showed efficacy and no cytotoxicity [70]. **K9CATH** is a 38-mer synthetic form of a canine cathelicidin originally found in neutrophil granule contents. K9CATH possesses salt-independent antimicrobial activity but loose activity in serum. It shows good in vitro efficiency against *C. albicans* [71] and in vivo, it has been evaluated for conditions caused by other microorganisms, such as *Staphylococcus aureus*-causing mastitis and pulmonary tuberculosis, both resulting in a significant decrease in microbial numbers and reduced signs of disease [72,73]. Peptide **ZY13** (15-mer) comes from a non-mammal cathelicidin, and is based on cathelicidin-BF from the venom of the snake *Bungarus fasciatus*. It shows improved serum stability, low hemolytic activity and cytotoxicity, and excellent antimicrobial activity against *C. albicans*. The effect of ZY13 was investigated in an inflammatory vaginitis mouse model induced by antifungal-resistant *C. albicans*, quickly clearing yeast cells and reducing infection-induced vaginal inflammation [74]. **Indolicidin** is a 13-mer cationic antimicrobial peptide isolated from bovine neutrophils. It does not acquire either an α-helical or β-domain structure. Indolicidin was studied in an animal model of disseminated candidiasis. Nanocomposites containing indolicidin conjugated to graphene oxide, a carbon-based nanomaterial used for drug delivery, successfully cleared *Candida* infection in the organs of immunocompromised mice and had low hemolytic activity [38]. Indolicidin has antibacterial and antifungal activities. Its mechanism of action has been studied in gram-negative bacteria. Its initial entry is through self-promoted uptake. Generally, in this pathway, AMPs displace cations such as Mg^2+^ and Ca^2+^ bound to negatively charged sites (ex., phosphate and/or carboxyl groups on LPS) [75]. However, in a lipid bilayer model system, indolicidin is bound to both neutral and anionic vesicles [76]. After the first interaction, indolicidin is observed to permeabilize both the outer and inner membranes, but it does not cause lysis. Furthermore, it appears that indolicidin uses its affinity for membrane components only to gain entry to the cytoplasm; once inside, it attacks other targets and causes the inhibition of DNA synthesis and promotes the filamentation of cells [77].

The three last peptides in the cathelicidin group, omiganan, protegrin-1, and iseganan, have fewer promising results, but there are valuable observations worth mentioning. **Omiganan** is a 12-mer synthetic peptide derived from indolicidin that has been subjected to several clinical trials. One outcome of these clinical trials is the reliable safety and tolerability of this peptide; however, evidence of its in vivo efficacy is still pending. In one study (NCT00231153), the capacity of omiganan to prevent catheter-related bloodstream infections, including candidemia, was evaluated; however, the results failed to show the efficacy of previous in vitro studies, in which the efficacy of omiganan against *Candida* spp. and other pathogens was evidenced [78]. **Protegrin-1** is an 18-mer β-sheet peptide isolated from porcine leukocytes. For its mode of action, it has been evidenced that protegrin-1 does not require metabolically active cells in order to exert its anti-microbial action [79]. Its initial cell entry follows more closely a barrel-stave model. Then, the pores that form allow negatively charged chloride ions to pass through, and then the cell countermeasures the change in the transmembrane potential by moving water molecules across the pore; this causes fast cell death by osmotic lysis [80]. Protegrin-1 shows good in vitro activity against *C. albicans* [81,82], but in preclinical studies, results are mixed. In a study that assessed a single injection of protegrin-1, the peptide was effective in increasing survival; however, in a model of subacute polymicrobial sepsis, it showed no improvement in survival, but microbial clearance was achieved [83,84]. Similarly, **iseganan**, a synthetic analogue of protegrin-1, has presented mixed results in clinical trials. Iseganan showed no efficacy in preventing the development of ulcerative oral mucositis, including oral candidiasis, in patients receiving chemotherapy [85,86], and in another study evaluating its efficacy in preventing ventilator-associated pneumonia, the iseganan treatment group had a higher mortality rate (NCT00118781).

**Histatins and mucins.** In mammalian salivary proteins, histatins and mucins can be found. Among AMPs, examples within these classes are as follows: histatin-5, its derived peptide PAC113, and MUC7 12-mer. Additionally, there is a chemokine with a sequence highly similar to histatin-5, CCL28. **Histatin-5** is a 24-mer peptide generated by the proteolytic cleavage of human histatin. Histatin-5 fungicidal activity is affected by the metal ions Mg^2+^ and Ca^2+^, but it does not depend on pH [87]. Histatin-5 binds to β-glucans and the Ssa1/Ssa2 Hsp70 chaperone in the cell wall of yeasts in an energy-independent manner, then its uptake is through an energy-dependent transporter-mediated process. Mutations that affect the involved transporters (Dur3 and Dur31) decrease susceptibility to histatin-5. Depending on the peptide concentration, other types of translocations across the membrane can happen: either direct transfer crossing and/or receptor-mediated endocytosis. Once inside the cell, histatin-5 affects multiple targets that lead to the cellular loss of ATP and to potassium ions generating an osmotic imbalance; then, delayed membrane lysis occurs [88]. An important effector of histatin-5 toxicity is Trk1p, which is the principal K^+^ transporter and the essential pathway for ATP loss [89]. Even though histatin-5 showed efficacy in a murine model of vulvovaginal candidiasis and for reducing the fungal burden in the vagina (up to 0.58 log_10_ colony-forming units (CFU)/mL) [47], a smaller fragment of histatin-5, **PAC113**, has been developed; this retains both antibacterial and antifungal activities. PAC113 (other names: PAC-113, P113 and P-113) is an α-helical 12-mer peptide. PAC113 cell entry is also facilitated by the Hsp70 chaperone, like its parent compound histatin-5. Inside *C. albicans* cells, it targets the mitochondrial complex I, increasing free radicals and inhibiting cellular respiration. It also has membrane-lytic activity [90,91]. The main focus of clinical trials using PAC113 is oral cavity health. In these clinical trials, PAC113 has proved to be safe, well-tolerated and effective against gingivitis, and Nal-P-113, a variant of PAC113 with 3 histidine residues replaced by β-naphthylalanine, improved chronic periodontitis [92,93,94]. However, in a clinical trial that was completed in 2008 (NCT00659971) and that assessed the effect of a PAC113 mouth-rinse in HIV seropositive individuals with oral candidiasis, there were no results available. While histatins are a family of histidine-rich peptides, mucins are large highly glycosylated proteins. **MUC7 12-mer** is derived from the mucin MUC7. First, a 20-mer peptide was designed based on MUC7, which showed potent antifungal activity against *C. albicans*, not dependent on cellular metabolic activity and not targeting mitochondria. Then, it was shown that MUC7 12-mer exceeded the antifungal activity of that 20-mer peptide [95]. An important characteristic of MUC7 12-mer is that its anti-microbial activity is enhanced by an alkaline condition, unlike other conventional drugs that lose their efficacy under high pH conditions [96]. MUC7 12-mer also showed efficacy in a murine model of oral candidiasis, with mice presenting oral thrush lesions. These successful results may be in part due to a favorable pH (6.0–7.2) and ionic strength (NaCl, about 40 mEq) prevailing in the oral cavity [51]. **CCL28** is a chemokine that is abundant in saliva. CCL28 improved the fungal burden of a murine model of oropharyngeal candidiasis with severe immunodeficiency (0.5 log). Given the sequence similarity between CCL28 and histatin-5, the study evaluated whether the sensitivity of *C. albicans* to CCL28 was affected if the same polyamine protein transporters (Dur3 and Dur31) that are required for histatin-5 translocation were missing, but the absence of the two transporters did not affect the *Candida*-killing activity. This suggests that CCL28 uses another membrane-crossing pathway [52].

**Antimicrobial peptides from milk and colostrum.** The other mammalian biological fluids abundant in AMPs are milk and colostrum. The representative peptide in these fluids is the iron-binding lactoferrin from which several peptides are derived: lactoferricin B, HLR1r, HLopt2, hLF1-11, and talactoferrin. **Lactoferricin B** consists of residues 17 to 41 of bovine lactoferrin. Lactoferricin B is affected by the presence of metal ions, Mg^2+^ and Ca^2+^, as is also observed for histatin-5. Interestingly, it was noted that these ions interact with certain cell constituents in a strain-dependent manner [97]. Contrary to histatin-5, lactoferricin B anti-*Candida* activity is pH-dependent. Antifungal activity has shown to be most effective around pH 5.0–6.0. In a murine model of oral candidiasis, in which the drinking water of the mice was adjusted to pH 5.5 and the peptide was administered by being dissolved in this water for three successive days, lactoferricin B reduced the total number of *Candida* cells in the oral cavity of mice even though there were no macroscopic changes in the tongue lesions. Meanwhile, a combination therapy with fluconazole significantly improved scores of tongue lesions [53]. **HLR1r** corresponds to positions 21–32 of human lactoferrin and has two modifications, a 6-residue arginine-rich motif designed to facilitate membrane interactions, and neutral capping at the N- and C-terminus. Its efficacy was demonstrated in a murine model of cutaneous candidiasis. A topical treatment of HLR1r (400 μg per spot) exhibited a 75% reduction in *C. albicans* cells in skin patches where the pathogen was applied [98]. Another lactoferrin-derived peptide, **Hlopt2**, had similar results in a murine *C. albicans* skin infection model with topical treatment (400 μg per spot), as a 72% decrease in the fungal load was observed [99]. Hlopt2 includes residues 20 to 31 of human lactoferrin, plus an additional cysteine at the N-terminus; in addition, Lys and Ala replace Gln24 and Asn26, respectively. Hlopt2 causes cytoplasmic and mitochondrial membrane permeabilization; deep pits are formed on the cell surface and *C. albicans* is killed rapidly. **hLF1-11** is composed of the first 11 residues of the N-terminal of human lactoferrin. In preclinical studies, hLF1-11 was effective in a mouse model of oral candidiasis and in a disseminated *C. albicans* infection in neutropenic mice. In this last study, it was also observed that hLF1-11 prevented the filamentous growth of the pathogen [39,100]. hLF1-11 has been the subject of one clinical trial that assessed the improvement of candidemia in patients, but it was withdrawn due to the unavailability of the patient population (NCT00509834). For its antimicrobial action, hLF1-11 requires metabolically active cells, and once inside the cell, it targets the mitochondria, causing an altered mitochondrial membrane potential, permeability and the release of ATP, the production of reactive oxygen radicals (ROS) and cell death. It also inhibits the yeast-to-hyphal switch, an established virulence trait in species of *Candida* [79,101]. It is possible that some peptides derived from lactoferrin require the Trk1p potassium transporter for fungicidal activity, but not the cell wall proteins SsA1/2p Hsp 70, as shown for histatin-5 and as was observed for a bovine lactoferricin-derived peptide-comprising residues 12 to 21 [102]. **Talactoferrin** is a recombinant human lactoferrin generated in *Aspergillus awamori* that only differs to human lactoferrin by its glycosylation due to the fungal system [103]. Talactoferrin initially had promising results in a clinical trial with patients with severe sepsis, including disseminated fungal infection. Enterally administered talactoferrin reduced mortality (12% absolute reduction in mortality) among patients with severe sepsis compared to the placebo treatment. Patients also received drotrecogin as the standard care for severe sepsis, at the discretion of the primary physician. Then, patients were contacted at 28 days, 3 months, and 6 months, in order to determine the outcome status; it was noted that a reduced mortality was maintained during this time period [104]. However, in another clinical trial (NCT01273779), there was a higher 28-day mortality rate in the talactoferrin group, and the study was terminated.

**Chromogranins.** Chromogranins are also proteins present in mammals. They are a family of acidic proteins abundant in neuroendocrine and immune cells, and secreted in biological fluids such as serum and saliva. Several AMPs, derived from the N-terminal region of chromogranin A, which was reported to have antimicrobial activity [105], are as follows: CGA-N9, CGA-N46, and CGA-N12. **CGA-N9** consists of residues 47 to 55. This peptide passes through the cell membrane via direct penetration and its internalization is energy-independent; it induces the depolarization of the cell membrane but does not prompt the formation of pores. It causes the translocation of calcium ions to the cytosol and mitochondria, disrupts the mitochondrial membrane potential, increases the levels of intracellular ROS, binds to DNA via an electrostatic interaction and finally, induces apoptosis in *C. tropicalis* [106,107]. In a preclinical study, CGA-N9 was administered for 14 days to a systemic candidiasis mouse model injected intraperitoneally with *C. tropicalis*. CGA-N9 increased the survival rate by 40%; an analysis of the total number of yeasts in the kidneys revealed a reduction of 97%, and significantly alleviation or elimination of the histopathological changes in major organs [40]. **CGA-N46** consists of amino acids 31 to 76 of the N-terminus of human chromagranin A. In a preclinical study, an immunocompromised mice model infected with *C. krusei* was administered a daily intraperitoneal injection of CGA-N46 for 2 weeks after 24 h of infection. CGA-N46 decreased the mortality (55.56% vs. 94.44%) and alleviated or eliminated histopathological symptoms in major organs [41]. A more effective derivative, **CGA-N12** (residues 65 to 76), which is more stable, has less hemolytic activity and a higher antifungal activity, was also obtained. Its mechanism of action is similar to the one described for CGA-N9. CGA-N12 also does not perturb the plasma membrane but dissipates the mitochondrial potential, increases levels of ROS, promotes Ca^2+^ uptake into the cytosol and mitochondria, fragments DNA, and finally causes apoptosis in *C. tropicalis* [108]. It has also been observed that CGA-N12 inhibits the activity of KRE9, which possesses β-1,6-glucanase activity, inhibiting cell wall synthesis [109]. CGA-N12 was evaluated in a disseminated *C. albicans* rabbit model, in which a reduced yeast load in the tissues, the suppression of inflammatory factors in the serum, and an improved histological outcome was observed. Another feature is that CGA-N12 inhibits the germ tube formation of *C. albicans* and *C. parapsilosis* [42]. CGA-N12 was evaluated in a similar preclinical study to the one carried out for CGA-N9. Mice were injected intraperitoneally with *C. tropicalis*, and a 14-day treatment with CGA-N12 followed. The highest dose reduced the load of yeast in the kidney by 99% and increased the survival rate by 45%; the histological outcome also improved [110].

#### 3.1.2. Antimicrobial Peptides from Fish

Among the AMPs derived from the peptides found in other vertebrates, there are two derived from fish, AP10W and Protamine peptide. **AP10W** is a 10-mer peptide derived from zebrafish AP-2 complex subunit mu-A, the heparin-binding protein, which is excreted by the neutrophils during sepsis. AP10W has two substitutions for tryptophan and one for isoleucine, in order to facilitate interaction with the lipid bilayer [111]. The mechanism of action of AP10W in fungi includes interacting with (1,3)-β-d-glucan, mannan and chitin in the cell wall, the depolarization of the cell membrane, the increment of ROS and finally apoptosis. AP10W was evaluated in a mouse wound model, in which the back skin of each mouse bearing an 8 mm in diameter wound was infected with *C. albicans*; one hour later, AP10W was applied to the wound. AP10W treatment promoted the closure and healing of skin wounds infected with the pathogenic yeast (up to 10% of the original size) and lowered yeast loads compared to the controls [56]. **Protamine** peptide is a 14-mer that comes from the arginine-rich protein protamine found in salmon spermatozoa. A cyclic variant with a disulfide bond, cyclic protamine peptide, showed higher antifungal activity and higher stability in high concentrations of NaCl. The cyclic form also had a higher efficacy in an immunosuppressed murine oral candidiasis model, which was treated with the peptide at 3, 24 and 27 h post-infection. The cyclic peptide also inhibited the hyphal formation of *C. albicans* on mouse tongues by 50%. The fungicidal mode of action includes peptide internalization by an energy-dependent mechanism, ATP efflux, and ROS generation [54].

### 3.2. Antimicrobial Peptides from Invertebrates

Antimicrobial peptides with anti-*Candida* activity that have their origin in invertebrates are obtained from members of the phyla Mollusca and Arthropoda. HG1 and phibilin can be found in mollusks and jelleine-I, melittin, cecropin A, lasioglossins III, gomesin, and GK-19 in arthropods.

#### 3.2.1. Antimicrobial Peptides from Mollusks

**HG1** is a homo-dimeric derivative of halocidin, isolated from the tunicate *Halocynthia aurantium;* each monomer contains 19 amino acid residues. This peptide is highly resistant to proteolytic degradation, and to the presence of fluid from human skin wounds, saliva, metal ions Ca^2+^ or Mg^2+^, and elevated salt concentrations [112]. HG1 interacts with (1,3)-β-d-glucan and forms ion channels within the membrane of *C. albicans* [113]. HG1 was investigated in a mouse model of oral candidiasis. Neutropenic mice were infected with *C. albicans* and treated with a solution of HG1, 3 days after infection, 3 times daily for 3 days. Tongues of HG1-treated mice had a significant improvement in symptoms (white patches), a favorable histological examination, and their candidacidal activity was rapid [55]. **Phibilin** is a 16-mer peptide identified in the mollusk *Philomycus bilineatus*. Phibilin disrupts the integrity of the plasma membrane, induces necrosis via the accumulation of ROS, inhibits the formation of biofilms and disrupts mature biofilms of *C. albicans*. Phibilin was investigated in a mouse cutaneous infection model, in which *C. albicans* was injected subcutaneously and 1 h later phibilin was directly injected subcutaneously into the infected area, once daily for 3 days. Phibilin significantly reduced skin abscesses and total yeast numbers [57].

#### 3.2.2. Antimicrobial Peptides from Arthropods

**Jelleine-I** is an 8-mer peptide derived from the major royal jelly (bee secretion) protein 1 precursor of the European honeybee *Apis mellifera*. Jelleine-I accumulates on the surface of membranes where it exerts pressure to accommodate its residues on the amphiphilic environment of the membrane; then, it forms pores and prompts cell lysis [114]. Jelleine-I was effective for a systemic infection caused by *C. albicans* and induced in neutropenic mice; the peptide was intraperitoneally injected once daily for 7 days. Low concentrations of the peptide (0.5 and 1 mg/kg) delayed death, while higher concentrations (5 and 10 mg/kg) significantly increased the survival of mice. Furthermore, jelleine-I demonstrated neglectable hemolytic activity and toxicity [43]. **Melittin** is a 26-mer lytic peptide also from *A. mellifera* but found in the honeybee venom. The pore-forming activity of melittin is best explained by the toroidal pore model, in which the polar section of the peptide interacts with the polar components of the lipid bilayer and remains in this orientation during the perpendicular transfer across the membrane [115]. Tetramers and octamers of melittin produce ring-like structures on the membrane. Melittin has a high affinity to chitin, and also inhibits (1,3)-β-d-glucan synthase, alters mitochondrial membrane potential, and causes an increase in ROS, phosphatidylserine externalization, DNA and nuclear fragmentation, and finally apoptosis [116]. Melittin has been the subject of preclinical studies for conditions caused by other etiological agents, in which it showed efficacy in eradicating the infection; meanwhile, no toxicity or hemolysis were observed [117,118]. Meanwhile, the in vitro evidence for melittin anti-*Candida* activity is extensive, and encompasses antifungal activity against *C. albicans*, *C. glabrata*, *C. krusei*, *C. tropicalis*, and *C. parapsilosis* [116]. **Cecropin A** is a 37-mer peptide from the giant silk moth, *Hyalophora cecropia*. Cecropin A has in vitro anti-*Candida* activity and displays low cytotoxicity against human cells. Cecropin A’s effects on *Candida* cells resemble those of melittin. It induces mitochondrial depolarization, caspase activation, phosphatidylserine externalization, DNA fragmentation, and apoptosis [119,120]. Cecropin A safety was also shown in a mouse model of inflammatory bowel disease, in which the peptide reduced inflammatory markers and enhanced beneficial gut microbiota [121]. **Lasioglossins** were discovered in the venom of the eusocial bee *Lasioglossum laticeps*. Lasioglossin III is a 15-mer peptide that prefers anionic lipids, permeabilizes the membrane in a non-disruptive manner, and may have intracellular targets [122,123]. Lasioglossin III was studied in a mouse model of induced vaginal candidiasis. Yeast infection was evaluated using a luciferase-producing *Candida* strain that enabled the measurement of total luminescence by whole body imaging. Infected zones decreased substantially at days 7 and 14 in the peptide-treated group, and the median CFU dropped up to 38% of the median CFU in the untreated group. [48]. **Gomesin** is an 18-mer cysteine-rich defense peptide isolated from the hemocytes of the tarantula spider *Acanthoscurria gomesiana*. Its disulfide bridges are essential for its antimicrobial activity and serum stability. Gomesin interacts with the lipid bilayer through electrostatic and hydrophobic interactions, which seem to be non-specific since the all-D enantiomer is as potent as the native molecule; this indicates the lack of a stereo-specific receptor. Most likely, gomesin interacts with the membrane in an orientation parallel to the membrane, then the hydrophobic residues are inserted into the membrane, and finally gomesin causes membrane permeabilization and death. Further experiments are needed to elucidate whether there is a formation of toroidal pores or of β-barrels when the peptide concentration is higher [124]. In a study on murine models of disseminated and vaginal candidiasis, gomesin was administered 1, 3 and 6 days after infection with *C. albicans*; the peptide significantly reduced the fungal burden of the kidneys, spleen, liver, and vagina [45]. **GK-19** is a 19-mer peptide derived from the venom of the African scorpion *Androctonus amoreuxi*. GK-19 was first derived from a peptide with a moderate antimicrobial activity that is highly hemolytic and toxic to mammalian cells, AamAp1. Through amino acid substitution, this peptide was improved. The effect of GK-19 on the membranes depends on the species of *Candida*. After GK-19 exposure, *C. krusei* had a slightly different membrane morphology than the one observed for *C. albicans* and *C. glabrata*; these last two species present deep cracks and ruptures on their membranes. A mouse model of scald combined with skin and soft tissue infections induced by *C. albicans* was used to evaluate GK-19 and its parent compound, AamAp1, in vivo. Four scalded wounds were made on mice backs and two days later these were infected with the pathogenic yeast. After day 10, it was observed that GK-19 significantly reduced the wound area (more than 95%) compared to the other treatments, and the histological examination revealed the full recovery of GK-19-treated mice, as their skin showed a large number of neovascularization and new-born hair follicle tissues. GK-19 also had a favorable toxicity/safety profile in vitro and in vivo [58].

### 3.3. Antimicrobial Peptides from Plants and Bacteria

There are approximately 368 AMPs classified in the Kingdom Plantae and 385 from bacteria [27]. Among these, **RsAFP2** is a 50-mer peptide found in radish seeds (*Raphanus sativus*) [125], which presents anti-*Candida* activity in vitro and was active in a prophylactic murine model of candidiasis. RsAFP2 interacts with fungal glucosylceramides, induces the mislocalization of septins, blocks the yeast-to-hypha transition, and induces apoptosis in *C. albicans* [126]. In the animal model, in which mice were inoculated intravenously with *C. albicans* and received the peptide 1 h after infection and at 24 h intervals for 4 days, RsAFP2 significantly reduced the fungal burden in the kidneys of infected mice, was not inactivated by serum, and was non-toxic to mammalian cells [44]. Among AMPs that are found in bacteria and have anti-*Candida* activity, cecropin-like N-terminal peptides derived from the ribosomal protein L1 (RpL1) of *Helicobacter pylori* can be mentioned; these are HP (2-20) and HPRP-A1. **HP (2-20)** is a 19-mer peptide that creates small pores (compared to melittin). In the fungal cell wall, it recognizes chitin but not chitosan or (1,3)-β-d-glucan, and can bind to neutral and anionic membranes. It has little or no hemolytic activity or cytotoxicity against mammalian cell lines [115]. The systemic antimycotic activity of HP (2-20) was tested in a lethal murine candidiasis model. In this study, it was shown that a high dose of the peptide (ca. 25 mg/kg body weight) and frequent intravenous administration (every 12 h) resulted in a 16.6% survival rate at day 14, while 92.5% of the untreated mice had died by day 3 [46]. **HPRP-A1** is a 15-mer α-helical peptide that exerts broad spectrum antimicrobial activity, including *Candida* species and HPRP-A2, which is an enantiomer of HPRP-A1 and is composed of all D-amino acids [127]. HPRP-A2 was tested in vivo in a mouse vaginitis model caused by *C. albicans* and was shown to inhibit 50% of the fungus [49].

### 3.4. Other AMPs

Other AMPs with anti-*Candida* activity are biologically inspired, such as XF-73 and WLBU2, but others do not resemble natural sources, such as CG_3_R_6_TAT, C(LLKK)_2_C, (LLKK)_3_C, and novamycin. **XF-73** (exeporfinium) is a dicationic porphyrin derivative with completed clinical trials, which just received a positive update from U. S. Food and Drug Administration on Phase 3, specifically for XF-73 nasal gel for the prevention of post-surgical staphylococcal infection (NCT03915470) [128]. The mechanism of action of XF-73 is very different from that of other AMPs. Similar to other AMPs, XF-73 affects cell membranes but does not cause lysis. However, XF-73 is a photosensitizer that exerts its cytotoxic effect after illumination; it associates with the mitochondrial membrane and cell death is due to the generation of ROS, in particular, singlet oxygen [129]. An in vitro study showed that XF-73 is a highly efficient photosensitizer of *C. albicans*. This treatment could be applied to superficial mucocutaneous mycoses, which is recurrent in immunodeficient patients. A 6 log_10_ reduction in *C. albicans* planktonic cells was achieved with a 15 min incubation with the peptide (0.5 µM) and irradiation with blue light; for biofilm cells, a longer incubation time (4 h) with 1 µM of XF-73 had a 5 log_10_ reduction [130]. **WLBU2** (PLG0206) is a 24-mer peptide derived from lentivirus lytic peptide 1. It was identified that the cytoplasmic tail of the transmembrane envelope protein of human immunodeficiency virus type 1 (HIV-1) had an amphipathic amino acid segment that is similar in structure but not in sequence to lytic AMPs [131]. WLBU2 not only shows anti-*Candida* activity in vitro, but it is also active against the yeast *Cryptococcus neoformans* [132], and is being evaluated in a Phase 1 clinical trial for treating periprosthetic joint infections (NCT05137314).

There is an increasing concern in the field that the introduction of AMPs in a general clinical use may accelerate the evolution of resistance to the natural defense peptides produced by the human immune system and thus compromise our natural defenses against infection [133]. This is the rationale behind the design of peptides with little resemblance to naturally encountered peptides. **CG_3_R_6_TAT** was designed as a short amphiphilic peptide containing a hydrophilic TAT segment, six arginine residues for adding a cationic charge, a hydrophobic segment of cholesterol and a spacer of three glycine moieties that separate the two distinct segments. Interestingly, this peptide can self-assemble into nanoparticles with the hydrophobic cholesterol at the core and the TAT shell arranged towards the surrounding environment. The nanoparticles can permeate the cell membrane due to their relatively large volume and they can also cross the blood–brain barrier [134,135]. CG_3_R_6_TAT was investigated in a rabbit model of candidal meningitis. The 11-day treatment with the peptide began 3 days after infection; after this, CG_3_R_6_TAT caused a significant reduction in fungal counts in the cerebrospinal fluid (CSF) of rabbits, reaching CSF sterilization at 8.5 days, and the histopathologic outcome was also significantly improved [60]. **C(LLKK)_2_C** and **(LLKK)_3_C** were designed to obtain peptides with α-helical folding with the following short recurring sequence: (XXYY)_n_, where X are hydrophobic residues, Y are cationic residues, n is the number of repeat units, and C is a cysteine substitution at the N- and/or C-terminals to improve antimicrobial activity (Table 1). The designed peptides exhibited antimicrobial activities against gram-positive bacteria and yeasts, and peptides with up to 3 repeat units had low MIC values and insignificant hemolysis [136]. These peptides were evaluated in a keratitis mouse model. Mice corneas were de-epithelialized and infected with an established *C. albicans* biofilm on a contact lens (2 mm). The peptide treatment was as follows: eye drops (20 µL each) were administered at 5-min intervals for the first hour, then every 30 min for the subsequent 7 h, and after 16 h every 1 h for another 8 h. Conventional antifungal agents (e.g., fluconazole) are no longer effective when a mature *C. albicans* biofilm is already established and they comprise a dense combination of yeasts, hyphae, and pseudohyphae. However, both designed peptides demonstrated strong antifungal activity against mature *C. albicans* biofilm, as 90% of yeasts were eradicated in 24 h after a single peptide treatment (500 mg/L for peptide (LLKK)_3_C, 1000 mg/L for peptide C(LLKK)_2_C). Visual examination showed less cloudy corneas and visible iris, and a histological examination revealed insignificant corneal epithelial erosion and no inflammation, immune cell infiltration, edema, or neovascularization [61]. **Novamycin** is a polyarginine molecule designed by NovaBiotics, which has been fully formulated as an antifungal drug candidate ready for clinical testing. Its intended use is to combat life threatening and drug-resistant bloodstream and tissue fungal infections, including those caused by *C. auris* [137]. Novamycin was obtained after investigating the effect of various cationic polypeptides; this peptide only contains arginine residues (Table 1). Novamycin kills *C. albicans* cells faster than caspofungin and retains activity in human whole blood and saliva. In vivo, it showed significant improvement in the fungal burden in the vaginal and in oropharyngeal candidiasis models [50].

## 4. Antimicrobial Peptides in Combination Therapies

Drugs can be used in combination to evaluate synergism (when the combined activity of two agents is higher than the added activity of each agent separately). Most of the information available for AMPs in combination therapies comes from in vitro studies. Even though this preliminary data is needed, it may lack correlation with the clinical outcome. However, clinical data of combination therapy are also poor, given several difficulties such as the low incidence and high mortality rates of fungal infections and the presence of comorbidities in patients, which increase mortality rates [138].

### 4.1. Echinocandins in Combination Therapies

Combinations of echinocandins with different drugs have been evaluated for *C. glabrata,* which has limited susceptibility to fluconazole and is increasing its echinocandin resistance. Caspofungin showed synergism with itraconazole, posaconazole, and amphotericin B. Among 17 isolates tested, between 17 and 29% presented synergism with these combinations [139]. In another study, two echinocandins, caspofungin and micafungin, showed synergism with nikkomycin Z. An echinocandin-resistant *C. albicans* strain presented a 64-fold and 4-fold decrease in the median minimum inhibitory concentration (MIC) for caspofungin and micafungin combined with nikkomycin Z, respectively. In addition, *C. parapsilosis* strains showed a median MIC reduction with these combinations, a 2- to 4-fold and 2- to 64-fold decrease for caspofungin and micafungin, respectively [140]. Colistin, a non-antifungal AMP, showed synergistic effect when combined with caspofungin but not with micafungin against *C. auris* [141]. In addition, combining caspofungin and posaconazole for treating a *C. albicans* infection in a murine systemic infection model showed a synergistic effect against some echinocandin-resistant *C. albicans* strains [142].

### 4.2. Lactoferrin-Derivatives in Combination Therapies

Several lactoferrin derivatives have shown synergy with other AMPs or conventional drugs. Lactoferrin was evaluated in combination therapy with fluconazole and itraconazole against ergosterol and pump *C. albicans* mutants. The peptide enhanced the anti-*Candida* activity of fluconazole only in the ergosterol mutants and of itraconazole in all strains. Lactoferrin did not influence the amount of intracellular fluconazole, indicating that it does not change the uptake of the drug [143]. In another study, the combination of lactoferrin with fluconazole, amphotericin B or 5-fluorocytosine was investigated against three *C. albicans*, four *C. glabrata*, and one *C. tropicalis* clinical isolates. It was observed that the most potent effect was obtained with the combination of lactoferrin and fluconazole. This effect was evident when lower amounts of the peptide were used. For example, a combination of 0.5 mg/mL of lactoferrin and 100 μg/mL of fluconazole resulted in 50% extra anti-*Candida* activity, meanwhile 25 mg/mL of lactoferrin and 3.3 μg/mL of fluconazole only achieved a 5% synergistic effect [144]. hLF1-11 showed synergy with caspofungin against caspofungin-resistant or -susceptible *C. albicans*, *C. parapsilosis*, and *C. glabrata* planktonic cells, and the inhibition of biofilm formation and the preformed biofilms of *C. albicans* and *C. parapsilosis* strains. The synergy between the two drugs restored sensitivity to caspofungin in caspofungin-resistant strains [145]. hLF1-11 also acted in synergy with fluconazole against *C. albicans*, *C. glabrata*, *C. krusei*, *C. parapsilosis*, and *C. tropicalis*. Interestingly, strains must be exposed to the peptide before the azole and no candidacidal activity was observed when the order was inverted; it was also observed that an antagonist of the ATP receptors completely blocked the effect of the combination [146]. Talactoferrin was combined with fluconazole and amphotericin B against *C. albicans* at multiple drug dose ratios, and the combinations had a synergistic effect, except for one talactoferrin:fluconazole ratio in one strain [147]. Combinations of lactoferricin B and fluconazole or itraconazole were examined against strains highly resistant to azoles. The use of sub-MIC of lactoferricin B in these combinations resulted in a significant decrease in the MICs of the other drugs, for example, the MIC of fluconazole for one strain changed from >256 to 0.25 μg/mL [148]. Another study evaluated lactoferricin B in combination with several antifungal drugs, and no synergism was observed between the peptide and amphotericin B, nystatin or 5-flucytosine, but a synergistic effect was evidenced with fluconazole, itraconazole, and ketoconazole [149]. In a murine model of oral candidiasis, the administration of lactoferricin B reduced the total number of *Candida* cells in the oral cavity of mice, but there were no macroscopic changes in the tongue lesions. Meanwhile a combination therapy with fluconazole significantly improved scores of tongue lesions [53].

### 4.3. Other AMPs in Combination Therapies

Combinations of LL-37 with fluconazole, amphotericin B or caspofungin have been investigated. The combination of LL-37 and fluconazole showed a synergistic effect in 80% of *C. auris* strains, and the rest of the combinations had a synergistic effect in all of the clinical strains of *C. auris* tested [150]. Omiganan has been observed to have an additive (fractional inhibitory concentration index, FICI, of 0.5–1.0) and/or synergistic (FICI ≤0.5) effect with fluconazole against clinical strains isolated from vulvovaginal candidiasis and bloodstream infections [151]. The salivary mucin-based peptide, MUC7 12-mer, was combined with PAC113 or miconazole against *C. albicans*, resulting in an additive effect; FICIs of 0.63 and 0.56 were observed, respectively. Results were synergistic also for *C. neoformans* [152]. The plant defensin-based peptides, RsAFP1 and RsAFP2, were also evaluated in combination with caspofungin and amphotericin B against *C. albicans* biofilms. At concentrations of 2.5 μg/mL to 10 μg/mL, RsAFP2 acted in synergism with caspofungin, preventing biofilm formation and destroying mature biofilms; no synergistic effects were observed against planktonic cells. Contrary to RsAFP2, RsAFP1 alone was not able to prevent or eradicate *C. albicans* biofilms, but presented the same synergistic effect when combined with caspofungin at the equivalent concentrations used for RsAFP2 [153]. A derivative of lasioglossin III, LL-III/43, prevented *C. albicans* biofilm formation in combination with clotrimazole. At concentrations of 25 µM for LL-III/43 and 50 µM for clotrimazole, the combination also had a significant effect on other virulence factors, as it abolished *C. albicans* hemolytic activity, and decreased the production of phospholipases and proteases by the pathogenic yeast. LL-III/43 alone prevented *C. albicans* biofilm formation on a material used in orthopedic surgeries (Ti-6Al-4 V alloy), but only the combination of LL-III/43 with clotrimazole prevented the formation of biofilms on urinary catheters [154]. Phibilin displayed in vitro synergistic effects with clotrimazole against *C. albicans*, but not with amphotericin B, nystatin or anidulafungin [57]. In a mouse model of *Candida* vaginitis, in which yeast biofilms were first established in mouse vaginas and then drugs were administered daily for 8 consecutive days, HPRP-A2 presented a synergistic effect when combined with chlorhexidine acetate. When lower doses of the drugs were used (0.5 mg/mL HPRP-A2 and 0.02 mg/mL chlorhexidine), they caused a significant reduction in the number of yeast cells recovered from the vagina of infected mice, approximately 30–40% when independently administered and 82% when combined. At higher doses (1.0 mg/mL HPRP-A2 and 0.5 mg/mL chlorhexidine), the drugs had an inhibition rate of 50% when separately administered, and 99.9% in combination [155].

## 5. Conclusions

In this review, we present a wide-ranging list of AMPs that have undergone preclinical or clinical trials with promising clinical applications, including fighting against major fungal infections. AMPs with anti-*Candida* activity that have been evaluated in different animal disease models, such as disseminated, vulvovaginal, oral, cutaneous and urinary tract candidiasis, as well as candidal meningitis and candidal keratitis, are revised. Generally, these AMPs present different structures including, α-helices, β-sheets, neither α-helical nor β-domain structure, and cyclical. Some exhibit similar cellular effects, such as the generation of ROS, membrane permeabilization, and ion and ATP leakage, while some others have different cellular targets that have not been unveiled. Importantly, there are several AMPs that recognize different cell membrane components; as such, if used in combination therapy, these AMPs can decrease the probability of membrane-related resistance mechanisms, for example, reduced cell wall negative charge. Overall, we believe that AMPs are avant-garde molecules that, alone or in combination therapies, can serve as alternative antifungal agents for the treatment of infections caused by medically important *Candida* species, which will strongly impact on diverse medical practices.

## 6. Future Directions

Antimicrobial peptides are good candidates for drug development even though some display good activity in vitro but poor activity in vivo. Nevertheless, the preferential interaction of several AMPs with specific fungal targets makes them non-toxic to mammalian cells, which is an essential requirement for any antimicrobial agent. Additionally, chemical adjustments and modifications, including shortening, cyclization, terminal changes, the addition of unnatural amino acids, residue phosphorylation, among others, can be utilized to increase the biological activity, effectiveness, absorption and stability of AMPs, as well as to reduce their toxicity. Furthermore, the use of AMPs in combination with other antimicrobial molecules or conventional therapy can result in synergistic interactions that help to increase the potency of the therapy while reducing therapeutic doses and the side effects of a particular drug when used alone. Therefore, it is very important to continue working on the development of strategies and approaches for the optimization of AMPs, as they are promising, viable and alternative treatments that can fight against fungal infections. With the threat of the emergence of antimicrobial resistance and considering the undesirable side effects of currently available antifungal medications, AMPs are excellent candidates as novel therapeutic agents.

## Figures and Tables

**Figure 1 pharmaceutics-15-00789-f001:**
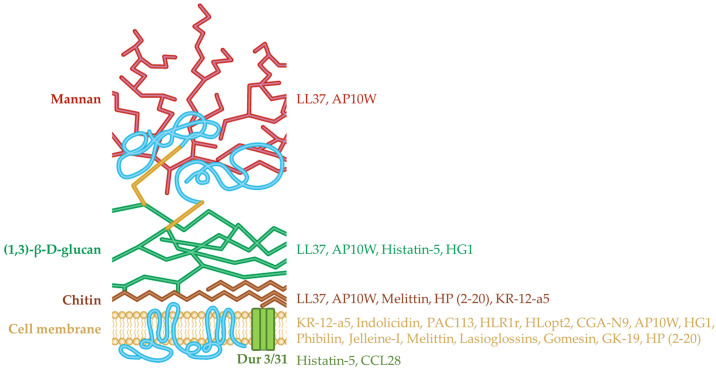
Schematic representation of cell membrane and cell wall targets of antimicrobial peptides with anti-*Candida* activity.

**Table 1 pharmaceutics-15-00789-t001:** Antimicrobial peptides that have been the subject of preclinical or clinical studies against infections caused by *Candida* species and that have been tested in vitro.

Family/Source	Name	Sequence	Length	*Candida* Species Tested In Vitro
Defensins	RTD-1	GFCRCLCRRGVCRCICTR	18-mer	*C. albicans, C. tropicalis*
	Bac8c	RLWVLWRR	8-mer	*C. albicans, C. parapsilosis*
Cathelicidins	LL-37	LLGDFFRKSKEKIGKEFKRIVQRIKDFLRNLVPRTES	37-mer	*C. albicans*
	KR-12-95	KRIVKLILKWLR	12-mer	*C. albicans*
	K9CATH	RLKELITTGGQKIGEKIRRIGQRIKDFFKNLQPREEKS	38-mer	*C. albicans*
	ZY13	VKRWKKWRWKWKKWV	15-mer	*C. albicans*
	Indolicidin	ILPWKWPWWPWRR	13-mer	*C. albicans*
	Omiganan	ILRWPWWPWRRK	12-mer	*C. albicans, C. glabrata*
	Protegrin-I	RGGRLCYCRRRFCVCVGR	18-mer	*C. albicans, C. tropicalis*
	Iseganan	RGGLCYCRGRFCVCVGR	17-mer	*C. albicans, C. glabrata, C. krusei*
Histatins and mucins	Histatin-5	DSHAKRHHGYKRKFHEKHHSHRGY	24-mer	*C. albicans, C. tropicalis, C. guillermondi*
	PAC113	AKRHHGYKRKFH	12-mer	*C. albicans, C. glabrata, C. krusei, C. parapsilosis, C. tropicalis*
	MUC7 12-mer	RKSYKCLHKRCR	12-mer	*C. albicans, C. glabrata*
	CCL28	HRKKHHGKRNSNRAHQGKHETYGHKTPY	28-mer	*C. albicans*
From milk and colostrum	Lactoferricin B	FKCRRWQWRMKKLGAPSITCVRRAF	25-mer	*C. albicans*
	HLR1r	FQWQRNMRKVRGSRRRRG	18-mer	*C. albicans*
	HLopt2	CFQWKRAMRKVR	12-mer	*C. albicans, C. glabrata, C. krusei*
	hLF1-11	GRRRRSVQWCA	11-mer	*C. albicans*
Chromogranins	CGA-N9	RILSILRHQ	9-mer	*C. glabrata, C. parapsilosis, C. krusei, C. tropicalis*
	CGA-N46	PMPVSQECFETLRGHERILSILRHQNLLKELQDLALQGAKERAHQQ	46-mer	*C. albicans, C. glabrata, C. parapsilosis, C. krusei, C. tropicalis*
	CGA-N12	ALQGAKERAHQQ	12-mer	*C. albicans, C. glabrata, C. parapsilosis, C. krusei, C. tropicalis*
From fish	AP10W	WKIKRWAIWK	10-mer	*C. albicans*
	Protamine	VSRRRRRRGGRRRR	14-mer	*C. albicans, C. glabrata, C. krusei, C. tropicalis*
From mollusks	HG1	KWLNALLHHGLNCAKGVLA	19-mer	*C. albicans, C. glabrata, C. krusei*
	Phibilin	RGDILKRWAGHFSKLL	16-mer	*C. albicans*
From arthropods	Jelleine-I	PFKLSLHL	8-mer	*C. albicans, C. glabrata, C. parapsilosis, C. krusei, C. tropicalis*
	Melittin	GIGAVLKVLTTGLPALISWIKRKRQQ	26-mer	*C. albicans, C. parapsilosis*
	Cecropin A	KWKLFKKIEKVGQNIRDGIIKAGPAVAVVGQATQIAK	37-mer	*C. albicans*
	Lasioglossin III	VNWKKILGKIIKVVK	15-mer	*C. albicans, C. parapsilosis, C. tropicalis*
	Gomesin	XCRRLCYKQRCVTYCRGR	18-mer	*C. albicans, C. glabrata, C. tropicalis*
	GK-19	GFLFKLIPKAIKKLISKFK	19-mer	*C. albicans, C. glabrata, C. krusei*
From plants	RsAFP2	PyroGlu-KLCQRPSGTWSGVCGNNNACKNQCIRLEKA RHGSCNYVFPAHKCICYFPC	50-mer	*C. albicans*
From bacteria	HP (2-20)	AKKVFKRLEKLFSKIQNDK	19-mer	*C. albicans*
	HPRP-A1	FKKLKKLFSKLWNWK	15-mer	*C. albicans*
Other AMPs	XF-73	KJLHJQGDBJNMPUXRFOENPRSAL	25-mer	*C. albicans*
	WLBU2	RRWVRRVRRWVRRVVRVVRRWVRR	24-mer	*Candida* spp.
	CG_3_R_6_TAT	Cholesterol-GGGRRRRRRYGRKKRRQRRR	20-mer	*C. albicans*
	C(LLKK)_2_C	LLKKLLKK	8-mer	*C. albicans*
	(LLKK)_3_C	LLKKLLKKLLKK	12-mer	*C. albicans*
	Novamycin	RRRRRRRRRRRRR	13-mer	*C. albicans, C. auris*

**Table 2 pharmaceutics-15-00789-t002:** Antimicrobial peptides (AMPs) that have been assessed in diverse animal models of *Candida* infection.

Disease Model	Animal	*Candida* Species	AMP (Reference)	Administration Route
Disseminated candidiasis	BALB/c mice	*C. albicans*	Indolicidin [38]	Intraperitoneal
	Swiss mice ^1^	*C. albicans* ^2^	hLF1-11 [39]	Intravenous
	BALB/c mice	*C. tropicalis*	CGA-N9 [40]	Intraperitoneal
	Kunming mice	*C. krusei*	CGA-N46 [41]	Intraperitoneal
	New Zealand white rabbits	*C. albicans*	CGA-N12 [42]	Intraperitoneal
	Kunming mice ^1^	*C. albicans*	Jelleine-I [43]	Intraperitoneal
	Mice	*C. albicans*	RsAFP2 [44]	Intravenous
	BALB/c mice	*C. albicans*	Gomesin [45]	Intraperitoneal
	CBA/J mice	*C. albicans*	HP (2-20) [46]	Intraperitoneal and intravenous
Vulvovaginal candidiasis	BALB/c mice	*C. albicans*	Histatin-5 [47]	Vaginal perfusion
	DBA/2 mice	*C. albicans*	Lasioglossin III [48]	Intravaginal
	BALB/c mice	*C. albicans*	Gomesin [45]	Vaginal cream
	ICR mice	*C. albicans*	HPRP-A1 [49]	Intravaginal solution
	BALB/c mice	*C. albicans*	Novamycin [50]	Vaginal solution
Oropharyngeal candidiasis	ICR mice	*C. albicans*	MUC7 12-mer [51]	Emulsion on the oral cavity
	C57BL/6J mice ^3^	*C. albicans*	CCL28 [52]	Oral gel
	ICR mice	*C. albicans*	Lactoferricin B [53]	Oral solution
	ICR mice	*C. albicans*	Protamine [54]	Oral solution
	ICR mice ^4^	*C. albicans*	HG1 [55]	Oral rinse
	CD1 mice	*C. albicans*	Novamycin [50]	Topical emulsion
Cutaneous candidiasis	ICR mice	*C. albicans*	AP10W [56]	Topical solution
	CD-4 mice	*C. albicans*	Phibilin [57]	Subcutaneous injection
	Kunming mice	*C. albicans*	GK-19 [58]	Topical
Urinary tract candidiasis	BALB/c mice	*C. albicans*	LL-37 [59]	Transurethral catheter
Candidal meningitis	New Zealand white rabbits	*C. albicans*	CG_3_R_6_TAT [60]	Intravenous
Candidal keratitis	C57BL/6 mice	*C. albicans*	C(LLKK)_2_C [61]	Eye drops
	C57BL/6 mice	*C. albicans*	(LLKK)_3_C [61]	Eye drops

^1^ Neutropenic mice. ^2^ Strain resistant to fluconazole. ^3^ Immunosuppressed with high-dose steroid treatment. ^4^ Immunosuppressed with cyclophosphamide.

**Table 3 pharmaceutics-15-00789-t003:** Antimicrobial peptides (AMPs) that have been assessed in clinical trials and are major therapeutically relevant classes.

Family/Source	Name	Clinical Trial ID	Condition or Disease	Phase	Formulation
Cathelicidins	LL-37	NCT04098562 NCT02225366	Diabetic foot ulcer Melanoma	Phase 2 Phase 1 and 2	Topical cream Intratumoral injection
	Omiganan	NCT00231153	Catheter infections	Phase 3	Topical gel around the catheter insertion site
	Iseganan	NCT00118781	Pneumonia	Phase 2 and 3	Topical application to the oral cavity
Histatins and mucins	PAC113	NCT00659971	Oral candidiasis	Phase 2	Mouth rinse
From milk and colostrum	hLF1-11	NCT00509834	Candidaemia	Phase 1 and 2	Intravenous bolus
	Talactoferrin	NCT01273779	Severe sepsis	Phase 2 and 3	Oral treatment
Other AMPs	XF-73	NCT03915470	Staphylococcal and surgical site infections	Phase 2	Nasal gel
	WLBU2	NCT05137314	Joint infection	Phase 1	Irrigation solution

## Data Availability

The data supporting this review are from previously reported studies and datasets, which have been cited. The processed data are available from the corresponding author upon request.

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
