# Peer review of "Antimicrobial Peptides: Avant-Garde Antifungal Agents to Fight against Medically Important Candida Species"

_pharmaceutics, 2023, doi:10.3390/pharmaceutics15030789_

Round 1
Reviewer 1 Report
In this review, " Antimicrobial peptides: avant-garde antifungal agents to fight 2
against medically important Candida species” Gina P. Rodríguez-Castaño et.al., has described a wide-ranging list of AMPs that have undergone pre-clinical or clinical trials with promising clinical applications, including fighting against major fungal infections. The authors also have exemplified AMPs with the anti-Candida activity that has been evaluated in different animal disease models.
The comments and suggestions for this manuscript are as follows-
1. The introduction and the main body of the manuscript are typical textbook types. This is lacking intellectual input from authors. The author should provide a comprehensive introduction, host-pathogen interaction of Candida albicans, and antimicrobial peptides. The author should avoid repetitive information, and if necessary, a large amount of text can be replaced with appropriate references.
2. The author may include the “mode of action of AMPs” on Candida albicans in table 1.
3. Figure 1, Tables 2 and 3 are looking good and informative.
Author Response
Comments and Suggestions for Authors
In this review, " Antimicrobial peptides: avant-garde antifungal agents to fight against medically important Candida species” Gina P. Rodríguez-Castaño et.al., has described a wide-ranging list of AMPs that have undergone pre-clinical or clinical trials with promising clinical applications, including fighting against major fungal infections. The authors also have exemplified AMPs with the anti-Candida activity that has been evaluated in different animal disease models.
Answer. Thanks to the reviewer for the time to read and evaluate the manuscript and for each of the valuable comments and suggestions. In the revised manuscript and along with this point-by-point response, we aimed to address all issues mentioned by this and other reviewers.
The comments and suggestions for this manuscript are as follows:
- The introduction and the main body of the manuscript are typical textbook types. This is lacking intellectual input from authors. The author should provide a comprehensive introduction, host-pathogen interaction of Candida albicans, and antimicrobial peptides. The author should avoid repetitive information, and if necessary, a large amount of text can be replaced with appropriate references.
Answer. Thanks to the reviewer for this suggestion. The interaction host-pathogen and AMPs were included in the introduction, as well as 2 new references. The whole text was also checked to avoid repetitive information. Considering that our manuscript is a “Review” we believe that we complied with the article type description stated at the author guidelines.
- The author may include the “mode of action of AMPs” on Candida albicans in table 1.
Answer. Thanks to the reviewer for this suggestion. Since most specific modes of action of AMPs against Candida species -and many other microorganisms- remain incompletely understood, we illustrate, in Figure 1, known targets that some of the peptides have with the cell membrane or with specific molecules of the cell wall. In addition, as most known mechanisms of AMPs have been described in bacteria, it is important to consider that these mechanisms differ from other microorganisms, like fungi, which are eukaryotes, as the antifungal action depends on specific protein targets, which many times are unknown. From the peptides included in this review (>40), the target located in the membrane and/or a cell wall component, has been described only in 20, and this is what we are depicting schematically in Figure 1. In this way, specific mechanisms for each of the peptides listed in Table 1 are not always known, as such, including the “mode of action” in some rows of the table but not in all, will make the table look incomplete. If, apart from the target in the membrane and/or a cell wall, additional intracellular targets or characteristics of the mechanism of action of determined AMP against Candida are known, these have been mentioned in the text (for example, pore formation, membrane disruption, apoptosis, etc).
- Figure 1, Tables 2 and 3 are looking good and informative.
Answer. Thanks to the reviewer for evaluating the figure and tables.
Reviewer 2 Report
.
Author Response
Thanks to the reviewer for the time to read and evaluate the manuscript.
Reviewer 3 Report
1.Line 105:Maybe you can change to a more suitable and detailed subtitle.
2.Line 105:This part is mainly about the antibacterial mechanism of antimicrobial peptides and the antimicrobial activity of antimicrobial peptides. Can you enumerate the antifungal mechanism of antimicrobial peptides and connect these two parts?
3.Line 146: ‘synergistic mixture of AMPs’ Does it refer to the combined use of different antimicrobial peptides or the combined use of antimicrobial peptides and existing antifungal antibiotics?
4.Line 166: This part may be added after the summary of the classification of antifungal peptides.
5. Generally, the published year of the reference are too early, please try to include more reference in the recent 5 years.
Author Response
Comments and Suggestions for Authors
Answer. Thanks to the reviewer for the time to read and evaluate the manuscript and for each of the valuable comments and suggestions. In the revised manuscript and along with this point-by-point response, we aimed to address all issues mentioned by this and other reviewers.
- Line 105:Maybe you can change to a more suitable and detailed subtitle.
Answer. Following the suggestion of the reviewer, the subtitle was changed.
- Line 105:This part is mainly about the antibacterial mechanism of antimicrobial peptides and the antimicrobial activity of antimicrobial peptides. Can you enumerate the antifungal mechanism of antimicrobial peptides and connect these two parts?
Answer. Following the suggestion of the reviewer, the antifungal mechanisms of AMPs were mentioned, and the two parts were connected.
- Line 146: ‘synergistic mixture of AMPs’ Does it refer to the combined use of different antimicrobial peptides or the combined use of antimicrobial peptides and existing antifungal antibiotics?
Answer. Thanks to the reviewer for noticing this. The term “synergistic mixture of AMPs” refers to the use of diverse peptides produced in the same organism. This is now stated in the text.
- Line 166: This part may be added after the summary of the classification of antifungal peptides.
Answer. Following the suggestion of the reviewer, the Figure 1 was added after the summary of the classification of antifungal peptides.
- Generally, the published year of the reference are too early, please try to include more reference in the recent 5 years.
Answer. Thanks to the reviewer for this suggestion. References were checked through the text. They include the most recent articles in which different AMPs have been tested or were described against Candida infections.
Reviewer 4 Report
The review by Rodríguez-Castaño et al is a comprehensive review regarding the role of AMPs as novel antifungals against Candida spp. infections. With antifungal resistance on the rise and the high incidence of candidosis, this review is not only informative but crucial to call the attention of the need for new antifungals. The text is fluid and well-written, and the images are clear. I recommend acceptance.
Author Response
Thanks to the reviewer for the time to read and evaluate the manuscript and for the valuable comments.